# Novel De Novo *BRCA2* Variant in an Early-Onset Ovarian Cancer Reveals a Unique Tumor Evolution Pathway

**DOI:** 10.3390/ijms26052295

**Published:** 2025-03-05

**Authors:** Gianmaria Miolo, Giovanni Canil, Maurizio Polano, Michele Dal Bo, Alessia Mondello, Antonio Palumbo, Fabio Puglisi, Giuseppe Corona

**Affiliations:** 1Medical Oncology and Cancer Prevention Unit, Centro di Riferimento Oncologico di Aviano (CRO), Istituto di Ricovero e Cura a Carattere Scientifico (IRCCS), 33081 Aviano, Italy; gmiolo@cro.it (G.M.); fabio.puglisi@cro.it (F.P.); 2Immunopathology and Cancer Biomarkers Unit, Centro di Riferimento Oncologico di Aviano (CRO), Istituto di Ricovero e Cura a Carattere Scientifico (IRCCS), 33081 Aviano, Italy; giovanni.canil@cro.it; 3Experimental and Clinical Pharmacology Unit, Centro di Riferimento Oncologico di Aviano (CRO), Istituto di Ricovero e Cura a Carattere Scientifico (IRCCS), 33081 Aviano, Italy; mpolano@cro.it (M.P.); mdalbo@cro.it (M.D.B.); alessia.mondello@cro.it (A.M.); 4Pathology Unit, Centro di Riferimento Oncologico di Aviano (CRO), Istituto di Ricovero e Cura a Carattere Scientifico (IRCCS), 33081 Aviano, Italy; antonio.palumbo@cro.it; 5Department of Medicine, University of Udine, 33100 Udine, Italy

**Keywords:** *BRCA2* gene, de novo variant, proteomics, collagen, MMR proteins, extracellular matrix

## Abstract

Ovarian cancer (OC) is a highly heterogeneous malignancy, often characterized by complex genomic alterations that drive tumor progression and therapy resistance. In this paper, we report a novel de novo *BRCA2* germline variant NM_000059.3:c.(8693_8695delinsGT) associated with early-onset OC that featured two regions with differential MMR (Mismatch Repair) gene expression. To date, only six cases of de novo *BRCA2* variants have been reported, none of which were associated with early-onset high-grade serous OC. The immunohistochemical analysis of MMR genes revealed two distinct tumor areas, separated by a clear topographic boundary, with the heterogeneous expression of MLH1 and PMS2 proteins. Seventy-five percent of the tumor tissue showed positivity, while the remaining 25% exhibited a complete absence of expression, underscoring the spatial variability in MMR gene expression within the tumor. Integrated comparative spatial genomic profiling identified several tumor features associated with the genetic variant as regions of loss of heterozygosity (LOH) that involved *BRCA2* and *MLH1* genes, along with a significantly higher mutational tumor burden in the tumor area that lacked MLH1 and PMS2 expression, indicating its further molecular evolution. The following variants were acquired: c.6572C>T in *NOTCH2*, c.1852C>T in *BCL6*, c.191A>T in *INHBA*, c.749C>T in *CUX1*, c.898C>A in *FANCG*, and c.1712G>C in *KDM6A*. Integrated comparative spatial proteomic profiles revealed defects in the DNA repair pathways, as well as significant alterations in the extracellular matrix (ECM). The differential expression of proteins involved in DNA repair, particularly those associated with MMR and Base Excision Repair (BER), highlights the critical role of defective repair mechanisms in driving genomic instability. Furthermore, ECM components, such as collagen isoforms, Fibrillin-1, EMILIN-1, Prolargin, and Lumican, were found to be highly expressed in the MLH1/PMS2-deficient tumor area, suggesting a connection between DNA repair deficiencies, ECM remodeling, and tumor progression. Thus, the identification of the *BRCA2* variant sheds light on the poorly understood interplay between DNA repair deficiencies and ECM remodeling in OC, providing new insights into their dual role in shaping tumor evolution and suggesting potential targets for novel therapeutic strategies.

## 1. Introduction

De novo mutation rates exhibit considerable variability across cancer predisposition syndromes, with conditions such as Neurofibromatosis type 1 (NF1) and Familial Adenomatous Polyposis (FAP) showing particularly high frequencies of these genetic events [1,2,3].

Conversely, Hereditary Breast and Ovarian Cancer (HBOC) syndrome demonstrates much lower rates of de novo mutations, reflecting a relatively more stable pattern of inheritance, as demonstrated in a comprehensive cohort study that included 12,805 consecutive unrelated patients diagnosed with breast cancer (BC) and/or ovarian cancer (OC), where only 0.4% of patients (3/801; CI 0.1–1.1%) carried a de novo *BRCA1* variant and 0.1% (1/726; CI 0.02–0.8%) harbored *BRCA2* de novo [4]. These findings underscore the minimal contribution of de novo variants to the overall incidence of HBOC while emphasizing the critical role of family history in genetic screening.

To date, overall, only 18 cases of de novo *BRCA1/2* variants have been reported, with 12 involving *BRCA1* [4,5,6,7,8,9,10,11,12,13] and 6 involving *BRCA2* [4,14,15,16,17,18]. Interestingly, these genetic variants were primarily identified in individuals with a median age of cancer onset that rarely exceeded 40 years. Among the reported cases, only one patient that carried a *BRCA1* variant was diagnosed with OC at the age of 39 [4], while none of the six patients with de novo *BRCA2* variants developed OC [4,14,15,16,17,18].

Establishing the de novo origin of a genetic variant is challenging due to the difficulty of obtaining parental DNA samples, particularly for patients with cancers that manifest later in life, which explains why most de novo variants are identified in individuals under the age of 40. This constraint also partially account for the relative rarity of OC cases linked to de novo *BRCA1/2* variants, as OC generally exhibits lower penetrance and occurs at a later median age than BC [19,20].

Deficiencies in DNA repair pathways can interact within specific tumor subsets, and their cross-talk may significantly influence the penetrance of pathogenic variants. MMR (Mismatch Repair) gene mutations impair mismatch correction, leading to base substitutions and compensatory BER (Base Excision Repair) pathway activation. NER (Nucleotide Excision Repair) and HR (Homologous Recombination) also cooperate, particularly in *BRCA1/2*-mutated tumors, where HR deficiency triggers compensatory NER activation. Similarly, MMR proteins influence HR stability, where MLH1 maintains chromosome integrity by stabilizing HR and preventing lesion accumulation [21,22,23,24,25,26].

In this context, we present the case of a woman diagnosed with OC at age 41 who carried a previously unreported de novo *BRCA2* variant NM_000059.3:c.(8693_8695delinsGT). To investigate whether additional genetic factors contributed to the penetrance of this variant, we performed an in-depth analysis of the tumor’s immunohistochemical (IHC), spatial genomic, and proteomic profiles, offering novel insights into the complex interplay between genetic predisposition and cancer development.

## 2. Results

### 2.1. Case Presentation

In September 2021, a 41-year-old woman that presented with dysuria underwent a transvaginal ultrasound, which revealed a suspected ovarian lesion. Based on this evidence and the elevated serum level of the CA125 marker at 411 U/mL (normal range 0.0–45.0 U/mL), a Computed Tomography (CT) scan was performed using volumetric acquisition after the intravenous injection of an iodinated contrast agent (Ioexol, 90 mL) on a Philips Healthcare CT scanner. The scan revealed a multicameral pelvic mass with multiple nodules on the diaphragm, subglissonian capsule, and left falciform ligament.

A gynecological examination confirmed the presence of a vaginal nodule adherent to the rectum, and further ultrasound imaging identified bilateral solid ovarian masses. The patient underwent the resection of large nodules from the right diaphragm and Morrison’s pouch, bilateral oophorectomy, extraperitoneal hysterectomy, radical omentectomy, appendectomy, and splenectomy. Despite the comprehensive surgical approach, the procedure was suboptimal, leaving a residual tumor of less than 5 mm on the right diaphragm. A histological examination revealed high-grade serous adenocarcinoma with solid and papillary growth patterns and areas of necrosis (TNM pT3c pNX pM1b and FIGO Stage IVB). Following the surgery, the CA125 marker returned to normal levels, and the patient underwent six cycles of carboplatin (5 AUC) and paclitaxel (175 mg/m^2^), followed by maintenance therapy with olaparib (300 mg twice daily). Family tree analysis of the proband revealed multiple cancer cases, such as OC in the daughter of the mother’s first cousin at age 45, lung cancer in a maternal first cousin at age 74, bladder cancer in the maternal grandfather at age 80, and leukemia diagnosed in the paternal aunt at age 63 (Figure 1). This family history of cancer, along with the early onset of the OC cancer, prompted the investigation of genetic variants that may be associated with the development of the disease. Since both parents of the proband were healthy at the age of 73 and given the high prevalence of pathogenic variants in the *BRCA1* and *BRCA2* genes in ovarian cancer, as well as their significant therapeutic implications, the initial molecular analysis focused on these two genes.

### 2.2. Blood Molecular Analysis Results

The molecular analysis of blood-derived DNA from the proband identified a germline *BRCA2* heterozygous pathogenic variant in exon 21, specifically NM_000059.3:c.(8693_8695delinsGT). This frameshift variant involves the deletion of three nucleotides (TGC) at position 8693–8695 and the insertion of two nucleotides (GT), leading to a premature stop codon and the production of a truncated protein p.(Leu2898Cysfs*11) (Figure 1). This variant is located within the DNA Binding Domain (DBD) of the *BRCA2* gene and it has been classified as pathogenic according American College of Medical Genetics (ACMG) guidelines. Indeed, this variant results in a frameshift that leads to a loss of function of the BRCA2 protein (PVS1); it is absent in control populations from the Exome Sequencing Project, 1000 Genomes Project, or Exome Aggregation Consortium databases (PM2); it has been identified as de novo in a patient without a family history of the variant (PS2).

Further genetic testing of the proband’s relatives revealed that neither parents nor the siblings carried this variant, while paternity was confirmed through microsatellite analysis. Establishing the de novo origin of the variant is extremely important, as it excludes the possibility of recurrence in family members, except for the proband’s offspring. The de novo occurrence of this variant combined with the unusually early onset of OC, prompted an investigation into potential factors that could explain its high penetrance.

### 2.3. IHC Tumor Tissue Results

After an initial IHC analysis showed negative p53 and positive PAX8 staining, the MMR protein expression was assessed due to its role in tumorigenesis and its interaction with BRCA-related HR deficiencies.

The analysis revealed the normal expression of MSH2 and MSH6 proteins throughout the tumor. In contrast, MLH1 expression exhibited a specific heterogeneity, where approximately 75% of the tumor tissue (A1) showed positivity and the remaining 25% (A2) displayed complete absence. These areas were clearly separated by a distinct topographic boundary. Similarly, PMS2 expression mirrored the topographic pattern observed for MLH1 (Figure 2).

This spatial variation in the MMR (Mismatch Repair) protein expression may have significant implications for tumor biology and sensitivity to therapies, such as immune checkpoint inhibitors or PARP inhibitors, which target DNA repair deficiencies. This led to further molecular characterization of the A2 region to gain deeper insights into the mechanisms that underlay this heterogeneity and its clinical relevance.

### 2.4. Genomic Tumor Tissue Results

The comparative Next-Generation Sequencing (NGS) analysis between normal tissue and the two distinct tumor areas (A1 and A2) revealed relevant molecular features other than the loss of heterozygosity (LOH) of *BRCA2* gene and the 3p21.31 chromosomal sub-band where the MLH1 gene is located (Table 1).

The MLH1- and PMS2-expressing tumor area (A1) exhibited somatic variants in *MDC1* (c.3557T>C, VAF 0.0213), *TP53* (c.670G>T, VAF 0.768), and *LZTR1* (c.353G>A and c.2317G>A; VAFs 0.0574 and 0.7586, respectively) (Table 2). These genetic variants were all confirmed in the tumor area deficient in MLH1/PMS2 protein expression (A2), which confirmed a shared tumor origin; however, this region exhibited a significantly higher tumor burden, indicating its further molecular evolution.

Indeed, the following variants were further acquired from the A2 area: c.6572C>T in *NOTCH2* (VAF 0.0222), c.1852C>T in *BCL6* (VAF 0.0212), c.191A>T in *INHBA* (VAF 0.1441), c.749C>T in *CUX1* (VAF 0.1036), c.898C>A in *FANCG* (VAF 0.0208), and c.1712G>C in *KDM6A* (VAF 0.0403) (Table 2).

Interestingly, among these new somatic mutations, the VAF of the c.353G>A variant in the *LZTR1* gene increased to 0.3819, along with the loss of the c.2317G>A variant (Table 2), indicating the biallelic involvement of the gene.

### 2.5. Proteomic Tumor Tissue Results

The proteomic expression profiles were not consistently detected across all the tissue samples but revealed marked differences in the protein expression between the two regions with distinct MLH1/PMS2 patterns.

The Principal Component Analysis (PCA) protein expression profiles from normal, A1, and A2 tissues revealed a clear separation between the normal tissue and tumor regions with differing MLH1/PMS2 protein expression, a finding further supported by the heatmap, which showed distinct clusters and unique expression profiles consistent with PCA (Figure 3).

Notably, the A2 tumor area exhibited significant reductions in proteins essential to the DNA replication pathway, including MCM3, MCM4, POLD1, RFC2, and PCNA, all of which are pivotal for maintaining the integrity of DNA replication. Moreover, a significant decrease also resulted in proteins associated with DNA repair and MMR pathways, such as MSH2 and FEN1, as well as in proteins associated with epigenetic and chromatin modification, including DNMT1, MTAP, NUP210, and H1-10. Conversely, this tumor region displayed a significant increase in proteins essential for the structure, stability, and function of the ECM, including Collagen alpha-1(VI) (COL6A1), alpha-2(VI) (COL6A2), alpha-3(VI) (COL6A3), alpha-1(XIV) (COL14A1), alpha-2(I) (COL1A2), and alpha-1(I) (COL1A1) chains, along with other key ECM components, such as Fibrillin-1, EMILIN-1, Prolargin, and Lumican (Figure 3).

The enrichment analysis strongly validated the involvement of biological pathways associated with cellular stability, metabolism, and stress response, with a particular emphasis on DNA maintenance and repair mechanisms, such as MMR, BER (Base Excision Repair), and DNA replication pathways, as well as ECM and cell interaction pathways (Figure 3).

The interactions between COL6A3, EMILIN1, and HSPG2 underscore the central involvement of the ECM, while proteins such as NASP, HIF1X, and MSH2, which were at the edges, suggest fewer direct interactions within this network (Figure 3).

## 3. Discussion

Determining the origin of *BRCA2* genetic variants is particularly challenging, especially in OC, where the high median age at diagnosis limits parental analysis. Here, we report a case of a 41-year-old woman diagnosed with high-grade serous carcinoma of ovarian origin and that carried a novel variant in the *BRCA2* gene: NM_000059.3:c.(8693_8695delinsGT) p. p.(Leu2898Cysfs*11). The early diagnosis facilitated molecular investigations in her parents, confirming its de novo origin. To date, only six cases of de novo *BRCA2* variants have been reported [4,14,15,16,17,18], with none of them associated with early-onset high-grade serous OC. This finding is particularly noteworthy given that the risk of OC in *BRCA2* mutation carriers by this age is less than 1%, although it remains 13.7 times higher than in the general population (https://ask2me.org; the last access was on 31 January 2025) [19].

The genetic variant identified in this study was located within the DNA-binding domain (DBD) of BRCA2, a region specifically involved in binding single-stranded DNA (ssDNA) and playing a pivotal role in the HR repair process [27,28]. The latter begins when replication protein A recognizes single-stranded overhangs at DNA breaks, with the BRCA2 protein facilitating repair by interacting with RAD51. Variants such as the c.8693_8695delinsGT, which impairs the Oligo Binding 2 (OB2) loop within the BRCA2’s DBD, disrupt RAD51 filament formation on ssDNA, thereby hindering the initiation of the HR repair process [29].

The OC in this *BRCA2* genetic variant carrier exhibited a unique histological profile and appeared homogeneous under hematoxylin and eosin staining, yet revealed distinct longitudinal spatial heterogeneity in MLH1 expression, as revealed by immunohistochemistry. Approximately 25% of the tumor showed a deficiency in MLH1/PMS2 expression, with this area being clearly demarcated from the MLH1/PMS2-proficient region (Figure 1).

Genomic analysis of the tumor tissue revealed a high allelic frequency (VAF~90%) of the BRCA2 variant c.8693_8695delinsGT, which is indicative of LOH. This, combined with the concurrent loss of MLH1 expression, suggests the accumulation of unresolved R-loops, which increases the replicative stress and promotes genomic instability [30,31,32].

Despite the homogeneous tumor presence of *BRCA2* LOH, the MLH1/PMS2-deficient region exhibited a higher mutation burden, potentially indicating a distinct tumor evolutionary trajectory.

Among the variants identified in the MLH1-deficient tumor area, the c.670G>T (VAF 76.8%) pathogenic mutation in the *TP53* gene introduces a premature stop codon, resulting in a truncated protein that disrupts the DNA damage response, thereby acting as a tumor driver. Additionally, the c.353G>A likely pathogenic variant (VAF 5% and 38%) in the *LZTR1* gene impairs RAS ubiquitination, leading to the deregulation of the RAS/MAPK pathway, which promotes proliferative signaling and contributes to subclonal tumor progression.

Although alterations in the *LZTR1* gene are rare in OC (1.7%), changes in its expression are observed in 25% of cases, with 16.5% showing downregulation. This loss of expression has been associated with tumor progression and drug resistance, particularly in response to therapies targeting chromatin histone acetylation and Meiotic Recombination 11 (MRE11) (https://cancer.sanger.ac.uk). The last access was on 31 January 2025.

Spatial proteomic analysis of MLH1/PMS2-proficient and -deficient tumor tissues revealed distinct expression profiles between the two areas, which also differed significantly from normal tissue (Figure 3). Significant shifts in tumor dynamics were observed, which particularly affected ECM components and proteins involved in DNA repair and replication. Specifically, collagen type I α-chains were upregulated over 29-fold in the MLH1/PMS2-deficient tumor area. Such expression changes are commonly seen in cancers such as breast, gastric, lung, and pancreatic cancers, where interstitial matrix remodeling increases tissue stiffness, promotes cellular invasiveness, and drives tumor progression [33,34,35,36,37,38]. In cisplatin-resistant OC cells, collagen type I upregulation enhances the survival and drug resistance by modulating the ERK1/2 and GSK3β pathways while reducing the oxidative stress [39,40]. Similarly, in pancreatic cancer, elevated collagen type I levels promote gemcitabine resistance by upregulating HMGA2, a protein involved in chromatin remodeling, cell proliferation, and DNA repair [41,42].

Furthermore, collagen-integrin interactions within the ECM activate intracellular signaling pathways that drive β-catenin translocation into the nucleus, promoting the expression of cell cycle regulators, such as cyclin D1 [43,44]. Similarly, FBN1, which was upregulated 24-fold in the MLH1/PMS2-deficient tissue, has been associated with reduced cisplatin sensitivity, heightened chemoresistance, and a poorer prognosis [45].

In addition to the ECM-driven mechanisms, the reduced expression of minichromosome maintenance (MCM) proteins observed in the MLH1/PMS2-deficient tumor area underscores a critical aspect of genomic instability. Indeed, these proteins, which are essential for initiating and elongating DNA replication [46], may significantly impair DNA damage signaling, further promoting tumor progression and resistance to DNA-damaging therapies, such as platinum derivatives.

In summary, the analysis of distinct MLH1/PMS2-expressing tumor regions from a patient with a novel de novo *BRCA2* variant provided valuable insights into the evolution of OC. This variant compromises the DNA repair pathway, a hallmark of *BRCA2*-related cancers, and appears to drive significant ECM remodeling. This highlights the critical interplay between genomic instability and the tumor microenvironment, which is exacerbated by the loss of MLH1 expression, resulting in increased replication stress and R-loop formation. However, it is worth noting that this study is based on a single OC case, which may strongly limit the applicability of the findings. Thus, the unique molecular and immunohistochemical characteristics observed in this study may not be representative of a broader patient population.

## 4. Materials and Methods

### 4.1. Blood Molecular Analysis

The patient was referred for the identification of the more common genetic variants associated with hereditary cancer via DNA analysis from peripheral blood lymphocytes. Genomic DNA was extracted from blood using the Maxwell^®^ CSC Genomic DNA Kit (Promega, Madison, WI, USA), following the manufacturer’s instructions. Individual coding exons and their surrounding intronic sequences (with a minimum extension of −14 bp at the 5′ end and +10 bp at the 3′ end) were selectively amplified under multiplex conditions using the Devyser BRCA CE-IVD Kit (Devyser AB, Årsta, Sweden). Targeted resequencing was performed using NGS technology on the Illumina MiSeq platform, with paired-end 150-cycle reads to ensure a minimum coverage depth of 100 reads. The library was prepared using the Illumina DNA Prep Kit (Illumina, San Diego, CA, USA), following the manufacturer’s protocol to ensure high-quality sequencing data. A bioinformatic analysis of the sequencing data was carried out using Amplicon Suite v3.6.0 software (smartSeq srl).

Variants of uncertain or pathogenetic significance were confirmed using an independent DNA aliquot via single PCR amplification and direct Sanger sequencing. The MLPA (Multiplex Ligation-dependent Probe Amplification) test was performed with the SALSA MLPA CE-IVD Kit P002-D1 BRCA2/CHEK2, and data analysis was conducted using Coffalyser.net software (MRC-Holland, Amsterdam, The Netherlands).

The reference sequences used for *BRCA1* were MANE Select NM_007294.4 (mRNA) and NG_005905.2 (gDNA), while for *BRCA2*, the reference sequences were MANE Select NM_000059.4 (mRNA) and NG_012772.1 (gDNA). PCR amplification and Sanger sequencing were performed using the BigDye™ Terminator v3.1 Cycle Sequencing Kit (Thermo Fisher Scientific, Waltham, MA, USA). Sequencing was performed on an ABI Prism 3500 Genetic Analyzer (Thermo Fisher Scientific, Waltham, MA, USA), and the resulting chromatograms were analyzed for mutation or variant identification. The analysis was also conducted on the parents and two brothers to detect the specific variant. The de novo origin of the variant was confirmed by microsatellite analysis.

CA125 was measured using the Siemens Immulite 2000 CA125 Immunoassay Kit (Siemens Healthineers, Erlangen, Germany).

### 4.2. IHC Tumor Profile Analysis

Besides the histological characterization, the tumor tissues were also analyzed by IHC staining for p53 and PAX8 markers. Moreover, given the prognostic implications of changes in the expression of MLH1 in individuals with *BRCA2*-mutant tumors [30,31], an IHC staining for MMR proteins was carried out on 4 μm thick formalin-fixed, paraffin-embedded (FFPE) sections with the monoclonal antibodies for MLH1 (clone M1, Ventana), MSH2 (clone G219-1129, Cell Marque, Rocklin, CA, USA), MSH6 (clone SP93, Ventana), and PMS2 (clone A16.4, BD Biosciences). The MMR-deficient (dMMR) status was defined as the complete absence of expression for at least one MMR protein in the tumor tissue, with stromal and inflammatory cells serving as positive internal controls.

### 4.3. FFPE Tumor Tissue Investigations

To elucidate the molecular mechanisms underlying the cancer development and progression, a comprehensive spatial analysis of the tumor’s genomic and proteomic profiles was conducted.

Tumor DNA was extracted from the FFPE samples from two longitudinal continuous portions (proficient (A1) and deficient (A2) MLH1 expressions) of the OC and from normal tissue using the AllPrep DNA/RNA FFPE Kit (Qiagen, Hilden, Germany). Sequencing was performed using the TruSight Oncology 500 DNA kit (TSO500, Illumina, San Diego, CA, USA), where a panel of 523 genes was targeted. Library preparation was carried out manually according to the manufacturer’s protocol, followed by NGS on the NextSeq 550 instrument (Illumina, San Diego, CA, USA), where eight libraries were processed per run. Only coding variants with a coverage ≥ 50 reads were considered eligible, with blacklist regions excluded. Small variants were extracted from the TSO500 pipeline, and the filtered data were compiled into a CombinedVariantOutput.tsv file for each sample.

The protein expression profiles of tumor sections A1 and A2, along with a control tissue, were evaluated using FFPE slides. Protein extraction from the FFPE samples was performed following Ostasiewicz et al. [47] with minor adjustments. Briefly, tissue slides were warmed at 60 °C for 10 min to soften the paraffin. Deparaffinization was performed using xylene and ethanol in progressively decreasing ethanol–water mixtures (100%, 80%, 50%, 30%, 15%, and 5% ethanol) to ensure complete removal of the paraffin. Proteins were extracted using a solution that contained 5% Sodium Dodecyl Sulfate and 0.1 M Tris (pH 8.2), followed by incubation at 40 °C for 60 min in a thermomixer. The samples underwent sequential sonication and heating steps at 65 °C, 80 °C, and 90 °C to enhance the protein solubilization. After centrifugation, protein concentrations in the supernatants were determined using the Bicinchoninic Acid (BCA) assay, and samples were stored at −80 °C for downstream analysis. For the proteomic preparation, 300 µg of protein supernatant was precipitated with acetone (10 volumes of 80:20 cold acetone/water). The protein pellet was washed with acetone, dried, and rehydrated with trifluoroacetic acid. Neutralization was performed with a Tris base (pH 11), and reduction and alkylation were carried out using Tris(2-carboxyethyl)phosphine and 2-chloroacetamide at 95 °C for 5 min [48].

The protein solution was diluted in water (1:5 ratio), and digestion was performed with trypsin in a 1:50 enzyme-to-protein ratio at 37 °C for 16 h. Following the digestion, peptides were acidified to 2% formic acid and desalted using StrataX columns according to the manufacturer’s protocol. Purified peptides were evaporated to dryness and reconstituted in 50 µL of 0.1% formic acid for the LC-MS analysis. Peptides were analyzed in triplicate using a Peptide BEH C18 column, 3.5 µm, 150 × 2.1 mm (Waters) on a Q Exactive Plus mass spectrometer that operated in Data Independent Acquisition (DIA) mode. Chromatographic separation employed a gradient elution from 1% to 35% acetonitrile over 90 min at a flow rate of 0.2 mL/min. Electrospray ionization (ESI) parameters included a spray voltage of 3500 V in positive mode, a capillary temperature of 325 °C, and auxiliary gas at 10. The mass range was set to 400–1000 *m*/*z*, with a resolution of 17,500. The RAW data were processed using DIANN [49] (v1.8.1) in library-free mode, with parameters set up for one missed cleavage, precursor lengths of 7–30 amino acids, carbamidomethylation of cysteine as a fixed modification, and a false discovery rate (FDR) of 1%. Label-Free Quantification (LFQ) mode was employed to quantify the protein expression.

### 4.4. Statistical Analysis

To compare the proteomic expression profiles across the three samples, two tumor regions with differing MLH1/PMS2 expressions (A1 and A2) and paired normal tissue, statistical and clustering methods were employed. Protein expression intensity levels were log-transformed and mean-scaled and applied to a PCA to visualize the sample variability and clustering patterns, with 95% confidence ellipses computed based on the mean and covariance of each group. Hierarchical clustering analysis (HCA) was performed on normalized protein expression data using a two-way clustering approach to group proteins and samples based on similarities in their expression profiles. A one-way analysis of variance (ANOVA) was employed to identify proteins with significant differential expressions across the sample groups (A1, A2, and normal tissue). Multiple testing corrections were applied to control the FDR, with the likelihood of type I errors minimized using the Benjamini–Hochberg procedure to adjust the p-values, for the identification of significantly altered proteins while maintaining an FDR threshold of 5% [50]. Protein fold change (FC) values were calculated to compare the protein expression levels between the MLH1/PMS2-expressing tumor area (A1) and the normal tissue, as well as between the MLH1/PMS2 deficient tumor area (A2) and the normal tissue, selecting the protein that showed an A2/A1 ratio ≥ 2 or ≤ 0.5 with an FDR below 5% for further analysis.

The significantly differentially expressed proteins identified through statistical analysis were further subjected to enrichment analysis conducted using the Kyoto Encyclopedia of Genes and Genomes (KEGG) via GO enrichment analysis on the ShinyGO 80 platform [51] and by a protein–protein interaction (PPI) network analysis using the Search Tool for the Retrieval of Interacting Genes/Proteins (STRING). To ensure the reliability of the enrichment findings, statistical significance was assessed using an FDR threshold of <0.05, as implemented in both ShinyGO 80 and STRING, to ensure the identification of robust and biologically relevant pathways.

## 5. Conclusions

Overall, these findings suggest that the loss of DNA repair capacity not only fosters genetic alterations but also contributes to ECM restructuring, which, in turn, supports tumor progression, invasiveness, and chemoresistance.

Expanding these investigations to a larger cohort of patients with *BRCA2* mutations could further validate these findings, revealing common evolutionary trajectories that tumors adopt under the selective pressures of their microenvironments. Understanding these patterns could shed light on the molecular processes underpinning chemoresistance and tumor aggressiveness, offering a dual perspective on how genetic mutations and microenvironmental factors converge to drive cancer progression. Such insights could pave the way for more effective therapeutic strategies that target both DNA repair deficiencies and ECM remodeling in *BRCA*-mutated OC.

## Figures and Tables

**Figure 1 ijms-26-02295-f001:**
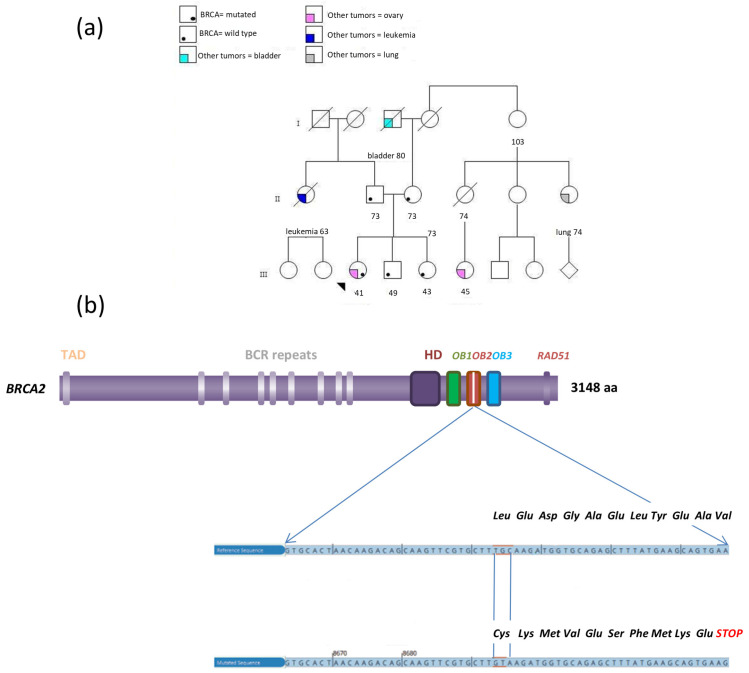
(**a**) Proband’s family tree analyzed for cancer history and genetic variant. Squares represent males, while circles represent females. Deceased individuals are marked with a diagonal line through the symbol. The number below each symbol indicates the age of the individual. (**b**) Structure of the BRCA2 protein. The eight BCR repeats and the three oligonucleotide/oligosaccharide-binding (OB) fold domains within the DNA Binding Domain (DBD) are highlighted. The OB2 domain features a distinct subregion called the Tower region, shown inwhite. Both the normal nucleotide and amino acid sequences are shown alongside their changes due to the variant.

**Figure 2 ijms-26-02295-f002:**
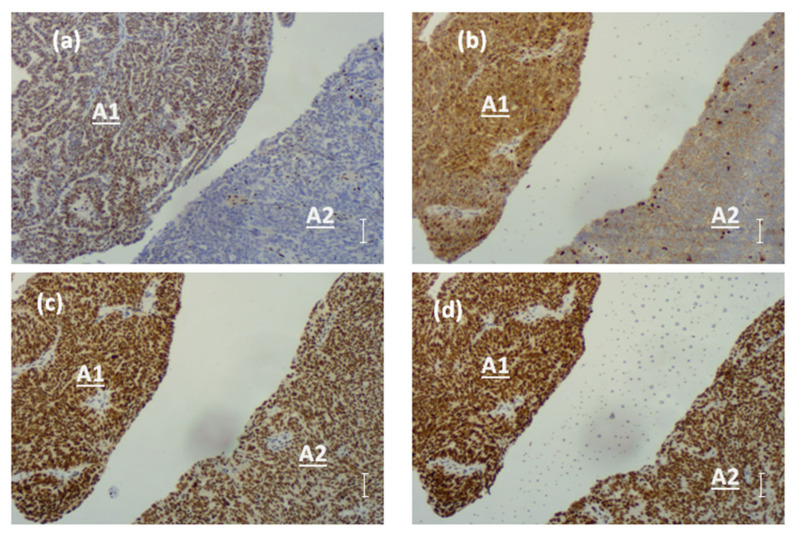
The IHC evaluation of MMR proteins highlighted their differential expression. MLH1 (**a**) and PMS2 (**b**) proteins were absent in area A2 compared with area A1, while MSH2 (**c**) and MSH6 (**d**) showed a consistent expression across the tumor. Staining observed in background lymphocytes and stromal cells represents the positive internal control. Scale bar: 100 µm; images captured at 100× magnification.

**Figure 3 ijms-26-02295-f003:**
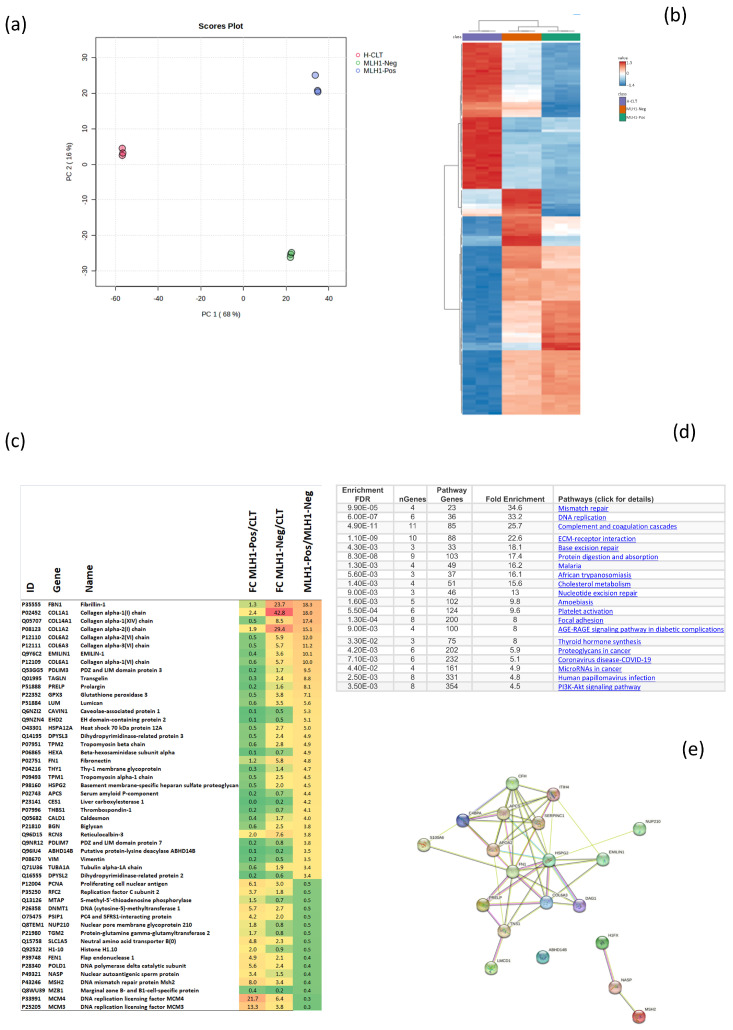
(**a**) PCA scores plot illustrating the separation of normal tissue (H-CLT), MLH1/PMS2-negative tumor tissue (A2), and MLH1/PMS2-positive tumor tissue (A1) along PC1 and PC2. (**b**) Heatmap illustrating the expression profiles of various genes across different tissue samples (H-CLT normal tissue, MLH1/PMS2-negative tumor tissue (A2), and MLH1/PMS2-positive tumor tissue (A1), with hierarchical clustering shown on both axes. The color scale represents the normalized expression values, with red indicating higher expression and blue indicating lower expression. (**c**) Fold change (FC) in gene expression between MLH1/PMS2-positive (A1), MLH1/PMS2-negative (A2), and H-CLT samples. Genes are listed with their corresponding IDs, gene names, and FC values, highlighting significant differences in expression levels across the samples. (**d**) Pathway enrichment using the Kyoto Encyclopedia of Genes and Genomes (KEGG) database. (**e**) Protein–protein interaction (PPI) network, generated using the STRING database.

**Table 1 ijms-26-02295-t001:** Summary of the genes and chromosomal regions that underwent a loss of heterozygosity (LOH). VAF—variant allelic frequency, N—normal, A1—tumor area 1, Sub-b—sub-band.

Gene	Position	VAF N	VAFA1	C-DOT Notation	Consequence	PMut	Band/sub-b
** *TNFRSF14* **	chr1	0.4506	0.1302	NM_003820.3:c.50A>G	Missense	p.(Lys17Arg)	1p36.12
** *TNFRSF14* **	chr1	0.4778	0.1472	NM_003820.3:c.721G>A	Missense	p.(Val241Ile)	
** *SPEN* **	chr1	0.4477	0.8781	NM_015001.2:c.3272T>C	Missense	p.(Leu1091Pro)	1p36.33
** *SPEN* **	chr1	0.4573	0.9002	NM_015001.2:c.7078A>G	Missense	p.(Asn2360Asp)	
** *SETD2* **	chr3	0.4884	0.8617	NM_014159.6:c.5885C>T	Missense	p.(Pro1962Leu)	3p21.31
** *MST1* **	chr3	0.5159	0.9046	NM_020998.3:c.2107C>T	Missense	p.(Arg703Cys)	3p21.3
** *MST1R* **	chr3	0.4579	0.0709	NM_002447.2:c.1301C>T	Missense	p.(Ser434Leu)	3p21.31
** *EPHA5* **	chr4	0.4718	0.1116	NM_001281765.2:c.242A>C	Missense	p.(Asn81Thr)	4q13.1
** *TET2* **	chr4	0.4849	0.9156	NM_001127208.2:c.5284A>G	Missense	p.(Ile1762Val)	4q24
** *FAT1* **	chr4	0.454	0.0924	NM_005245.3:c.10660T>G	Missense	p.(Ser3554Ala)	
** *FAT1* **	chr4	0.4722	0.0826	NM_005245.3:c.8798A>C	Missense	p.(Gln2933Pro)	4q35.2
** *FAT1* **	chr4	0.4985	0.0804	NM_005245.3:c.8152A>G	Missense	p.(Ile2718Val)	
** *FAT1* **	chr4	0.5222	0.0908	NM_005245.3:c.3818A>G	Missense	p.(His1273Arg)	
** *FAT1* **	chr4	0.511	0.9015	NM_005245.3:c.3190A>G	Missense	p.(Arg1064Gly)	
** *FAT1* **	chr4	0.5005	0.9108	NM_005245.3:c.1842C>G	Missense	p.(Phe614Leu)	
** *FAT1* **	chr4	0.509	0.8889	NM_005245.3:c.1444G>A	Missense	p.(Val482Ile)	
** *FAT1* **	chr4	0.4503	0.9089	NM_005245.3:c.1212T>G	Missense	p.(Ser404Arg)	
** *FAT1* **	chr4	0.4657	0.9182	NM_005245.3:c.1212T>G	Missense	p.(Ser404Arg)	
** *SDHA* **	chr5	0.4218	0.1061	NM_004168.3:c.1969G>A	Missense	p.(Val657Ile)	5p15.33
** *MAP3K1* **	chr5	0.5116	0.8961	NM_005921.1:c.2416G>A	Missense	p.(Asp806Asn)	5q11.2
** *MAP3K1* **	chr5	0.4807	0.9173	NM_005921.1:c.2716G>A	Missense	p.(Val906Ile)	
** *MSH3* **	chr5	0.4279	0.8582	NM_002439.4:c.235A>G	Missense: splice region	p.(Ile79Val)	5q14.1
** *MSH3* **	chr5	0.4507	0.8909	NM_002439.4:c.3133G>A	Missense: splice region	p.(Ala1045Thr)	
** *APC* **	chr5	0.4735	0.0957	NM_000038.5:c.5465T>A	Missense	p.(Val1822Asp)	5q22.2
** *FGFR4* **	chr5	0.4817	0.1022	NM_002011.4:c.28G>A	Missense	p.(Val10Ile)	5q35
** *PRKN* **	chr6	0.4715	0.8791	NM_004562.2:c.1138G>C	Missense	p.(Val380Leu)	6q26
** *NBN* **	chr8	0.4751	0.853	NM_002485.4:c.553G>C	Missense	p.(Glu185Gln)	8q21.11
** *PTCH1* **	chr9	0.5091	0.9231	NM_000264.3:c.3944C>T	Missense	p.(Pro1315Leu)	9q22.3
** *ABL1* **	chr9	0.5	0.0899	NM_007313.2:c.2972C>T	Missense	p.(Ser991Leu)	9q34.1
** *NOTCH1* **	chr9	0.4369	0.8764	NM_017617.4:c.2734C>T	Missense	p.(Arg912Trp)	9q34.3
** *TET1* **	chr10	0.4909	0.8843	NM_030625.2:c.485A>G	Missense	p.(Asp162Gly)	10q26.2
** *LATS2* **	chr13	0.5388	0.1168	NM_014572.2:c.971C>T	Missense	p.(Ala324Val)	13q12.11
** *LATS2* **	chr13	0.4629	0.1194	NM_014572.2:c.608C>T	Missense	p.(Ala203Val)	
** *BRCA2* **	chr13	0.4699	0.1175	NM_000059.3:c.1114A>C	Missense	p.(Asn372His)	13q13.1
** *BRCA2* **	chr13	0.444	0.8963	NM_000059.3:c.8693_8695delinsGT	Frameshift	p.(Leu2898Cysfs*11)	
** *MGA* **	chr15	0.4887	0.8672	NM_001164273.1:c.2146A>T	Missense	p.(Thr716Ser)	15q25.1
** *ANKRD11* **	chr16	0.4833	0.1178	NM_013275.5:c.2912C>T	Missense	p.(Ala971Val)	16q12.2
** *NCOR1* **	chr17	0.4835	0.0859	NM_006311.3:c.6544G>A	Missense	p.(Ala2182Thr)	17q24
** *ERBB2* **	chr17	0.4816	0.9158	NM_004448.3:c.1963A>G	Missense	p.(Ile655Val)	17q12
** *RNF43* **	chr17	0.499	0.0896	NM_017763.5:c.1252C>A	Missense	p.(Leu418Met)	17q21.31
** *RNF43* **	chr17	0.4607	0.1016	NM_017763.5:c.1028G>A	Missense	p.(Arg343His)	
** *RNF43* **	chr17	0.3561	0.8328	NM_017763.5:c.350G>A	Missense	p.(Arg117His)	
** *BRIP1* **	chr17	0.4582	0.1061	NM_032043.2:c.2755T>C	Missense	p.(Ser919Pro)	17q22
** *DOTL1* **	chr19	0.4835	0.1069	NM_032482.2:c.4156G>A	Missense	p.(Gly1386Ser)	19p13.12
** *PTPRS* **	chr19	0.4587	0.8987	NM_002850.3:c.2785C>T	Missense	p.(Arg929Cys)	19p13.2
** *TMPRSS2* **	chr21	0.4375	0.8995	NM_001135099.1:c.23G>T	Missense	p.(Gly8Val)	21q22.3
** *EP300* **	chr22	0.5123	0.1179	NM_001429.3:c.2989A>G	Missense	p.(Ile997Val)	22q13.2

**Table 2 ijms-26-02295-t002:** Summary of the gene mutations identified in different MLH1/PMS2-expressing tumor regions. It highlights the variant allelic frequency (VAF) for each mutation, along with the corresponding nucleotide changes (C-DOT notation) and predicted protein consequences (pMut).

Gene	Position	VAF A1	VAF A2	C-DOT Notation	Consequence	pMut
** *NOTCH2* **	chr1		0.0222	NM_024408.3:c.6572C>T	Missense variant	p.(Ala2191Val)
** *BCL6* **	chr3		0.0212	NM_001706.4:c.1852C>T	Missense variant	p.(Arg618Cys)
** *MDC1* **	chr6	0.0213	0.0242	NM_014641.2:c.3557T>C	Missense variant	p.(Val1186Ala)
** *INHBA* **	chr7		0.1441	NM_002192.3:c.191A>T	Missense variant	p.(Asn64Ile)
** *CUX1* **	chr7		0.1036	NM_181552.3:c.749C>T	Missense variant	p.(Ala250Val)
** *FANCG* **	chr9		0.0208	NM_004629.1:c.898C>A	Missense variant	p.(Leu300Met)
** *TP53* **	chr17	0.768	0.3527	NM_000546.5:c.670G>T	Stop gained: splice region	p.(Glu224Ter)
** *LZTR1* **	chr22	0.0574	0.3819	NM_006767.3:c.353G>A	Missense variant	p.(Arg118His)
** *LZTR1* **	chr22	0.7586		NM_006767.3:c.2317G>A	Missense variant	p.(Val773Met)
** *KDM6A* **	chrX		0.0403	NM_001291415.1:c.1712G>C	Missense variant	p.(Arg571Pro)

## Data Availability

The data that support the findings of this study are available in this manuscript.

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
