# Peer review of "Novel De Novo BRCA2 Variant in an Early-Onset Ovarian Cancer Reveals a Unique Tumor Evolution Pathway"

_ijms, 2025, doi:10.3390/ijms26052295_

Round 1

Reviewer 1 Report

Comments and Suggestions for Authors

Manuscript ID: ijms-3491478

Title: " Novel De Novo BRCA2 Variant Linked to Early-Onset Ovarian Cancer Unveils a Unique Tumor Evolution Pathway”.

 Authors : Miolo et al .

Manuscript Type:  Article

The manuscript of Gianmaria Miolo and colleagues reports a study aimed to describe the first case of a woman diagnosed with OC at age 41 who carried a previously unreported de novo BRCA2 variant. The authors  have  performed an  in-depth analysis of the tumor’s immunohistochemical (IHC), spatial genomic, and proteomic profiles, offering novel insights into the complex interplay between genetic predisposition and cancer development,  but however some specific criticisms should be addressed.

1) The lack of family history may be related to a de novo mutation, but in this case the proband revealed multiple cases of cancer, can you speculate on the presence of other mutations in other genes? Authors should  discuss  this aspect

2) The background section should discuss previous research which this study builds on. Several previous studies have assessed preliminary evidence of functional cross-linking and interaction between different DNA repair deficiencies in small subsets of tumors . However, none is

mentioned here

3) Genetic testing for a pathogenic variant in offspring of a carrier of a de novo variant is useful in identifying individuals at high risk for malignancies. Authors should mention the cascade test in the manuscript.

Author Response

Point by point response to REVIEWER 1

The lack of family history may be related to a de novo mutation, but in this case the proband revealed multiple cases of cancer, can you speculate on the presence of other mutations in other genes? Authors should discuss this aspect

Thank you for your insightful comment. We agree with the reviewer that the absence of a family history could suggest a de novo mutation, and the presence of multiple cancer cases in the proband raises the possibility of additional mutations in other genes. However, since both parents of the proband were healthy at the age of 73, and given the high prevalence of pathogenic variants in the BRCA1 and BRCA2 genes in ovarian cancer, as well as their significant therapeutic implications, the initial molecular analysis focused on these two genes. We will address this aspect in Section 2.1, Case Presentation.(Page 3 , lines 118-121)

The background section should discuss previous research which this study builds on. Several previous studies have assessed preliminary evidence of functional cross-linking and interaction between different DNA repair deficiencies in small subsets of tumors. However, none is mentioned here.

Thank you for this suggestion. We acknowledge the importance of discussing previous research that forms the basis of this study. We will incorporate relevant studies that have explored functional cross-linking and interactions between different DNA repair deficiencies in tumor subsets to provide a more comprehensive background. We introduce in the introduction section the following paragraph that include six new references: “Deficiencies in DNA repair pathways can interact within specific tumor subsets, and their cross-linking may significantly influence the penetrance of pathogenic variants. MMR (Mismatch Repair) gene mutations impair mismatch correction, leading to base substitutions and compensatory BER (Base Excision Repair) pathway activation. NER (Nucleotide Excision Repair) and HR (Homologous Recombination) also cooperate, particularly in BRCA1/2-mutated tumors, where HR deficiency triggers compensatory NER activation. Similarly, MMR proteins influence HR stability, with MLH1 maintaining chromosome integrity by stabilizing HR and preventing lesion accumulation [21-26]” .( Page 2-3 , lines 76-84)

Genetic testing for a pathogenic variant in offspring of a carrier of a de novo variant is useful in identifying individuals at high risk for malignancies. Authors should mention the cascade test in the manuscript.

Thank you for your valuable comment. We agree that genetic testing for a pathogenic variant in the offspring of a de novo variant carrier is essential for identifying individuals at high risk for malignancies. We ensure that now the manuscript explicitly mentions cascade testing to emphasize its importance by introducing in Section 2.2, Blood Molecular Analysis Results the following statement: “Establishing the de novo origin of the variant is extremely important as it excludes the possibility of recurrence in family members, except for the proband’s offspring” .( Page 5 , line 151 -153)

Reviewer 2 Report

Comments and Suggestions for Authors

Novel De Novo BRCA2 Variant Linked to Early-Onset Ovarian Cancer Unveils a Unique Tumor Evolution Pathway

International Journal of Laboratory Hematology

Title

Recommendation: Clearly indicate the type of study in the title (e.g., “case description”).

Abstract

Recommendation: The abstract is well written but only presents descriptive results without any data. Include data results (e.g., the assays performed and the results obtained).

For instance, the abstract does not mention the immunohistochemical (IHC), spatial genomic, and proteomic profiles.

Additionally, any statement claiming primacy (i.e., a claim of being the first to report the finding) should be suppressed.

Suggestion:
Include the identified variant using HGVS nomenclature.

Introduction

Recommendation: Suppress any statement claiming primacy (i.e., the claim of being the first to report the finding).

Results

Recommendation: Include the CA125 marker value and any other measurements when cited in the text.

Recommendation (lines 107–111): Use the HGVS nomenclature to describe genetic variants throughout the article, including the corresponding transcript. Verify variant names for correctness using tools such as Mutalyzer (https://mutalyzer.nl/normalizer).

Recommendation:mIn Table 1, the differentiation between tumor region A and tumor region B (as described in lines 145–148) is unclear. Define “VAF N” and “VAF T.”

Recommendation: Consider including the somatic variant classification in the tables, as this classification is currently mentioned only in the Discussion.

Recommendation: Classify the variant NM_000059.3 c.8693_8695delinsGT using ACMG rules and include both the classification and the rules applied in the manuscript.

Discussion

Recommendation: Some results (e.g., the c.670G>T (VAF 76.8%) pathogenic mutation in the TP53 gene) are described only in the Discussion. Ensure that all results are presented in the Results section and then appropriately discussed.

Recommendation:Comment on the limitations of the study.

Methods

Recommendation: On line 295, note that the SALSA MLPA PROBE MIX ME011 is not a DNA extraction kit; adjust the text accordingly.

Recommendation:Cite the manufacturer and country of origin for the Devyser BRCA CE-IVD kit. Extend this detail to include the other kits mentioned in the Methods section.

Recommendation: Describe the characteristics of the sequencing in terms of read type and read length (e.g., paired-end 150 cycles) and include the sequencing kit’s version and manufacturer.

Recommendation: Detail how direct Sanger sequencing was performed.

Recommendation:Clarify how each piece of data was obtained—for example, the CA125 marker and Computed Tomography (CT) data.

Recommendation: Provide evidence that formal ethical committee approval for the case description is not required in the patient’s/author’s country.

Figures and Figure Legends

Recommendation: The legend for Figure 1 is in Italian. Provide an English version.

Author Response

Point-by-point response to REVIEWER 2

Recommendation: Clearly indicate the type of study in the title (e.g., “case description”).

We have taken into account this recommendation and we will revise the title to make it clearer that this is a case description of a single patient by changing the title as follow: "Novel De Novo BRCA2 Variant in an Early-Onset Ovarian Cancer reveals Unique Tumor Evolution Pathway." ( Page 1 , lines  1-3)

Abstract

Recommendation: The abstract is well written but only presents descriptive results without any data. Include data results (e.g., the assays performed and the results obtained). For instance, the abstract does not mention the immunohistochemical (IHC), spatial genomic, and proteomic profiles.

Thank you for your valuable feedback. Following your suggestion the new version of the abstract now incorporates results from immunohistochemical (IHC), spatial genomic, and proteomic analysis in order to provide a more appropriate description of the main results. According, we introduce the following new sentences in the text of the abstract: “The immunohistochemical analysis of MMR genes revealed two distinct tumor areas, separated by a clear topographic boundary, with heterogeneous expression of MLH1 and PMS2 proteins. Seventy-five percent of the tumor tissue showed positivity, while the remaining 25% exhibited complete absence of expression, underscoring the spatial variability in MMR gene expression within the tumor. Integrated comparative spatial genomic profiling identified several tumor features associated to the genetic variant as regions of loss of heterozygosity (LOH), involving BRCA2 and MLH1 genes, along with a significantly higher mutational tumor burden in the tumor area lacking MLH1 and PMS2 expression, indicating its further molecular evolution. The following variants were acquired: c.6572C>T in NOTCH2, c.1852C>T in BCL6, c.191A>T in INHBA, c.749C>T in CUX1, c.898C>A in FANCG, and c.1712G>C in KDM6A.

Integrated comparative spatial proteomic profiles revealed defects in DNA repair pathways, as well as significant alterations in the extracellular matrix (ECM)”. ( Page 1-2 , lines 25-38)

Additionally, any statement claiming primacy (i.e., a claim of being the first to report the finding) should be suppressed.

Thank you for your suggestion. We recognize the importance of avoiding claims of primacy and we revised the manuscript to ensure that no such kind of statements are made by presenting the data and its implications objectively. ( Page 1 , lines 22-24)

Suggestion: Include the identified variant using HGVS nomenclature.

Thank you for your comment. More details regarding the identified variant has been now included in the revised version of the  abstract and the main text by adopting the HGVS nomenclature: NM_000059.3:c.(8693_8695delinsGT). ( Page 1 , line 20.)

Introduction

Recommendation: Suppress any statement claiming primacy (i.e., the claim of being the first to report the finding).

As mentioned above we revised the manuscript to ensure that no such kind of statements are made by presenting the data and its implications objectively. ( Page 3 , line 85)

Results

Recommendation: Include the CA125 marker value and any other measurements when cited in the text.

Following your recommendation we report in the result section the CA125 marker value. ( Page 3 , line 96.)

Recommendation (lines 107–111): Use the HGVS nomenclature to describe genetic variants throughout the article, including the corresponding transcript. Verify variant names for correctness using tools such as Mutalyzer (https://mutalyzer.nl/normalizer).

In the revision version o the manuscript we ensured that all genetic variants and the corresponding transcript are described using HGVS nomenclature as suggested. Moreover, we have taken into account the use of Mutalyzer tool for  the classification of the variant however we retain that the nomenclature more appropriate according also with the HGVS is NM_000059.3 c.8693_8695delinsGT.

Recommendation: Table 1, the differentiation between tumor region A and tumor region B (as described in lines 145–148) is unclear. Define “VAF N” and “VAF T.”

We revised the Genomic Tumor Tissue results section according with your observation. In particular we modified the legend of table 1 in order to better distinguish the tumor region. Additionally, we better defined  allelic frequency of the variant in normal and tumor tissues identified by the terms  "VAF N" and "VAF T ( Page 7 , line 187.)

Recommendation: Consider including the somatic variant classification in the tables, as this classification is currently mentioned only in the Discussion.

Thank you for your suggestion. We included the somatic variant classification also in the tables since it is an important aspect of providing a more clear context for the readers. ( inside the table 1)

Recommendation: Classify the variant NM_000059.3 c.8693_8695delinsGT using ACMG rules and include both the classification and the rules applied in the manuscript.

According to the reviewer suggestion the variant NM_000059.3 c.8693_8695delinsGT has been classified as pathogenic according to ACMG rules since it results in a frameshift, leading to a loss of function of the BRCA2 protein (PVS1); it is absent from controls in Exome Sequencing Project, 1000 Genomes Project, or Exome Aggregation Consortium databases (PM2); it has been identified as de novo in a patient without a family history of the mutation (PS2). ( Page 4 , lines 140 -146 .)

Discussion

Recommendation: Some results (e.g., the c.670G>T (VAF 76.8%) pathogenic mutation in the TP53 gene) are described only in the Discussion. Ensure that all results are presented in the Results section and then appropriately discussed.

We are sorry for this lack. In the revised version of the results section of the manuscript we included  the c.670G>T (VAF 76.8%) pathogenic mutation of the TP53 gene revealed in both areas of the tumor tissues. Moreover we ensured that all the results are reported before being discussed. ( Page 7 , line 190.)

Recommendation: Comment on the limitations of the study.

In the revised manuscript, we have better underlined the limitation of the study introducing in the discussion section the following paragraph : “This highlights the critical interplay between genomic instability and the tumor microenvironment, which is exacerbated by the loss of MLH1 expression, resulting in increased replication stress and R-loop formation. However, it is worth noting that this study is based on a single OC case which may strongly limit the applicability of the findings. Thus, the unique molecular and immunohistochemical characteristics observed in this study may not be representative of a broader patient population”. ( Page 11 , lines 324-330).

Methods

Recommendation: On line 295, note that the SALSA MLPA PROBE MIX ME011 is not a DNA extraction kit; adjust the text accordingly.

We are very sorry for the inaccurate reporting. In the revised version o the manuscript we correct report the DNA extraction kit used as following: “Genomic DNA was extracted from blood using the Maxwell® CSC Genomic DNA Kit (Promega, Madison, WI, USA) following the manufacturer's instructions”. ( Page 12 , lines 336-337 .)

Recommendation: Cite the manufacturer and country of origin for the Devyser BRCA CE-IVD kit. Extend this detail to include the other kits mentioned in the Methods section.

Following the recommendation of review we implemented the material and method section in other to include more details  about  the Devyser BRCA CE-IVD kit, as well as for the other kits used in the study. ( Page 12 , line 340)

Recommendation: Describe the characteristics of the sequencing in terms of read type and read length (e.g., paired-end 150 cycles) and include the sequencing kit’s version and manufacturer.

As above, we further revised the material and methods sections in order to include the characteristics of the sequencing, specifying the read type and read length as indicated by review. ( Page 12 , lines 340-345.)

Recommendation: Detail how direct Sanger sequencing was performed.

A more detailed methods description of the direct Sanger sequencing analysis is now reported in the revised manuscript . ( Page  , lines 356-360)

Recommendation: Clarify how each piece of data was obtained—for example, the CA125 marker and Computed Tomography (CT) data.

Following the recommendation of the review  we better report the methodology  used to obtain the CA125 marker and Computed Tomography (CT) data as following : “The determination of CA125 was performed using the Siemens Immulite 2000 CA125 Immunoassay Kit (Siemens Healthineers, Erlangen, Germany) (Page 12, lines 363-364). The CT scan was performed using volumetric acquisition after the intravenous injection of an iodinated contrast agent (Ioexol, 90 ml) on a Philips Healthcare CT scanner Erlangen, Germany”. ( Page , lines 97-101.)

Recommendation: Provide evidence that formal ethical committee approval for the case description is not required in the patient’s/author’s country.

All the patients provided informed consent for the execution of molecular investigations and for the publication of research findings. According to the country institutional ethical regulations this case study does not required a formally ethical committee approval.

Figures and Figure Legends

Recommendation: The legend for Figure 1 is in Italian. Provide an English version.

Thank you for your comment. We will provide an English version of the legend for Figure 1.
